# The Performance of Nonwoven PLLA Scaffolds of Different Thickness for Stem Cells Seeding and Implantation

**DOI:** 10.3390/polym14204352

**Published:** 2022-10-15

**Authors:** Timur Kh. Tenchurin, Alla V. Rodina, Vladimir P. Saprykin, Lada V. Gorshkova, Alexey A. Mikhutkin, Roman A. Kamyshinsky, Dmitry S. Yakovlev, Alexander L. Vasiliev, Sergey N. Chvalun, Timofey E. Grigoriev

**Affiliations:** 1National Research Centre “Kurchatov Institute”, 123098 Moscow, Russia; 2Natural Sciences Department, Moscow Region State University, 105005 Moscow, Russia; 3Shubnikov Institute of Crystallography of FSRC “Crystallography and Photonics” RAS, 119333 Moscow, Russia; 4Russian Quantum Center, Skolkovo, 121205 Moscow, Russia; 5Institute of Nano-, Bio-, Information, Cognitive and Socio-Humanistic Sciences and Technologies, Moscow Institute of Physics and Technology, State University, 141707 Dolgoprudny, Russia

**Keywords:** tissue engineering, fibrous scaffolds, polylactide, mesenchymal stem cells, bone marrow, microscopy

## Abstract

The 3D reconstruction of 100 μm- and 600 μm-thick fibrous poly-L/L-lactide scaffolds was performed by confocal laser scanning microscopy and supported by scanning electron microscopy and showed that the density of the fibers on the side adjacent to the electrode is higher, which can affect cell diffusion, while the pore size is generally the same. Bone marrow mesenchymal stem cells cultured in a 600 μm-thick scaffold formed colonies and produced conditions for cell differentiation. An in vitro study of stem cells after 7 days revealed that cell proliferation and hepatocyte growth factor release in the 600 μm-thick scaffold were higher than in the 100 μm-thick scaffold. An in vivo study of scaffolds with and without stem cells implanted subcutaneously onto the backs of recipient mice was carried out to test their biodegradation and biocompatibility over a 0–3-week period. The cells seeded onto the 600 μm-thick scaffold promoted significant neovascularization in vivo. After 3 weeks, a significant number of donor cells persisted only on the inside of the 600 μm-thick scaffold. Thus, the use of bulkier matrices allows to prolong the effect of secretion of growth factors by stem cells during implantation. These 600 μm-thick scaffolds could potentially be utilized to repair and regenerate injuries with stem cell co-culture for vascularization of implant.

## 1. Introduction

In recent years, many technologies have been developed that yield a variety of three-dimensional scaffolds, thus providing the foundation of biomaterials for tissue engineering as well as for in situ tissue regeneration and repair. To date, the basic requirements for biomaterial design have been defined; biomaterials should not only support the healing site but should also be bioactive and possibly biodegradable and should also influence cell behavior in a defined manner at the molecular level [1,2].

While biomaterials may consist of either natural or artificial components/materials, synthetic polymers have proven advantageous because they are easy to shape, they are tough, and they have high tensile properties. The mechanical and biochemical qualities and the degradation rates of synthetic biomaterials are easier to modify than those of natural polymers. They can be fabricated with specific spatial characteristics and precise mechanical properties. Poly-L/L-lactide (PLLA) has been widely explored as a scaffold biomaterial because of its biocompatibility, biodegradability, and excellent mechanical properties [3,4,5]. The biocompatibility of scaffolds is determined not only by their composition, but also by free surface energy, electrostatic interaction, surface roughness, and pore and fiber sizes [6,7,8,9]. Cell adhesion to nanofiber scaffolds (fiber diameter < 1 µm) is higher than the adhesion to microfiber ones (fiber diameter > 1 µm). However, smaller fiber diameter is often accompanied by insufficient scaffold pore size, preventing the cells from penetrating the scaffold [10]. On the other hand, while increased size of pores prevents the initial cell adhesion, it promotes the spreading of the cells and their differentiation. Changes in the fiber roughness do not significantly affect initial cell adhesion or differentiation, but can regulate the area of cell spreading. Besides, a significant drawback of PLLA scaffolds is their high hydrophobicity due to the lack of polar functional groups, which significantly limits their potential use in the clinic because the penetration of nutrient media into the structure of three-dimensional scaffolds is significantly hindered. Therefore, the efficiency of cell population growth in such scaffolds is rather low. By now, it is already clear that increasing porosity helps to overcome this problem, but there is no consensus on the ideal microstructure of such matrices: porosity, pore size, and fiber diameter. It is worth noting that although an increase in porosity and pore size leads to more efficient passage of cells into three-dimensional scaffolds, this is accompanied by a decrease in the mechanical strength of the materials. Therefore, in our work, highly porous matrices with high mechanical strength were obtained and the possibility of tissue integration into such matrices with different spatial organizations was studied to compare their effectiveness.

Synthetic surfaces seeded by endogenous cells following implantation usually possess an imposed design and microstructure based on the fabrication processes, which are perhaps uncharacterized or unknowingly derived from the manufacturing methods [11]. Numerous reports indicate that the scaffold microstructure significantly modulates cellular behavior [12,13,14,15,16,17]. In particular, it was found that the thickness of the scaffolds influences cell proliferation [17]. 

Mesenchymal stem cells (MSCs) are considered to be the cell type of choice for cell-based tissue engineering because of the ease with which they can be isolated and expanded, their multilineage differentiation capabilities, and their secretion of growth factors, such as vascular endothelial growth factor (VEGF), transforming growth factor-β (TGF-β), and hepatocyte growth factor (HGF), which act as chemoattractants [18]. In this regard, two global approaches can be distinguished for effective regeneration of damaged tissue using MSC transplantation on polymer matrices. Firstly, it is the transplantation of MSCs to replace the defect with the development of new cells from the implant and their directed differentiation. Secondly, stimulation of the migration of the body’s own cells to the implant area and an increase in their population as a result of the action of growth factors secreted by transplanted MSCs [19].

It has become clear that modulation of MSC function induced by the microstructural features of the scaffolds plays a central role in the regulation of cell behavior, which can be utilized in the fabrication of implantable scaffolds to enhance their biocompatibility. However, the in vivo characterization of PLLA scaffolds with various morphologies and the corresponding cell responses are still rather limited and that was mentioned in a review by Capuana et al. [20].

To facilitate our study of the cellular response to various scaffolds in vitro and in vivo, we used PLLA scaffolds, which we previously demonstrated to be promising for effective MSC growth [21]. The focus of this work was to compare the effect of the microstructural design of three-dimensional fibrous PLLA scaffolds on cell adhesion and proliferation and to examine scaffold biocompatibility and biodegradation after in vivo implantation with and without MSCs. 

Scaffolds thicknesses of 100 µm and 600 µm were selected based on the diffusional limit of oxygen required for cellular metabolism (100–200 µm) [22]. Thicker scaffolds have higher mechanical strength and are, in many cases, preferable as implants [23]. The fabrication of PLLA scaffolds with a thickness of more than 200 µm and an extracellular matrix-like microstructure has a wider range of potential applications: for reconstruction of complex tendon tears [24], auricular cartilage regeneration [25], and for vascular and vascularized tissue biofabrication [26]. However, a scaffold with acceptable functional properties in vitro may cause undesired effects in vivo since the latter includes all the biological and physical stimuli that dynamically change during the phases of tissue repair, for example, formation of new blood vessels becomes essential for a tissue to grow beyond the diffusion limit.

Therefore, to understand the influence of the scaffold microstructure on the integration of cells and their functional activity, two types of scaffolds were chosen, one with a thickness of 100 μm (close to the diffusional limit) and one that was considerably thicker (600 μm). These are the minimum and maximum possible thicknesses of the nonwoven materials. The lower limit of 100 μm is due to the minimum material strength and its ability to accumulate cellular structures. The upper limit is determined by the technological limitations of the electrospinning method. Preliminary study and our recent investigations of 400 μm PLLA nonwoven scaffold [27] demonstrated minor influence of thickness on the microstructure, thus in the present work, we chose markedly different scaffolds. It should be noted that relatively bulky scaffolds with a thickness of 600 μm could potentially be utilized to repair and regenerate injuries with stem cell co-culture for vascularization of implant.

It was also important to evaluate how long MSCs will remain in the composition of such matrices and how these matrices will affect the integration of endogenous cells.

## 2. Materials and Methods

### 2.1. PLLA Scaffolds

Two types of microfibrous scaffolds were fabricated from PLLA (molecular weight (MW) of 158 kDa) by electrospinning from a PLLA solution consisting of 8.6% polylactide, 82% chloroform, and 9.4% ethanol as previously described by Rodina et al. [28]. The matrices were produced by electrospinning on an experimental single-needle device developed at the Kurchatov Institute (Russia). For the electrospinning, a rotating metal electrode with a diameter of 63.6 mm was used at a rotation speed of 22 rpm. The PLLA solution was extruded through a spinneret using a syringe pump DS-08 (Visma-Planar, Minsk, Belarus) with a resistance of 294 Pa, which was connected to a Spellman SL130PN30 (USA) high voltage power supply. The extrusion rate of the solution during fiber formation was 10 mL/h. The distance between the needle tip and the collector electrode was 26–27 cm, and the voltage was at a range between 13 and 14.0 kV. A rhodamine 6G with a concentration of 1.5 × 10^−5^ g/g was added to the PLLA solution to enhance fiber contrast during the 3D study of the scaffold microstructure by confocal laser scanning microscopy (CLSM). Two types of scaffolds with thicknesses of 100 µm and 600 µm were fabricated and the formation time was 9 and 44 minutes, respectively.

### 2.2. Scanning Electron Microscopy

The PLLA scaffold microstructure and cell morphology were studied using a scanning electron microscope (SEM) Versa 3D DualBeam instrument (Thermo Fisher Scientific, Hillsboro, OR, USA) in low and high vacuum modes. Images were obtained in the secondary electron mode with accelerating voltages of 1 and 2 kV. These experimental conditions enhanced the topographic contrast on the surface of the PLLA fibers and attached cells, significantly reduced the charge accumulation, and improved the spatial resolution. Cross-sections of the scaffolds were prepared by cooling in liquid N_2_, followed by cleavage. Small cuts were made in the scaffold tissue to obtain a flat section in the desired place.

### 2.3. 3D Characterization of of the Scaffold: CLSM 

While BET is the commonly used integral specific surface area measurement method that is very sensitive on nanoscale, it is not always applicable in the analysis of microporous objects [29] and can give overestimated results [30]. Here, for the 3D characterization of the scaffolds and analysis of its pore sizes, the combination of direct CLSM and SEM methods was used. Experimental CLSM images constituting 3D data stacks were obtained on an Olympus FV10i-W CLSM (Olympus Corporation, Tokyo, Japan) using a laser with a wavelength of 473 nm and a UPLSAP60xW 60× water immersion objective lens with a numerical aperture of 1.2. The minimum aperture size for this microscope was used. The signal was recorded in the range of 490–590 nm. The pixel resolution of the CLSM images (X × Y) was 1024 × 1024 pixels and the lateral size of the images was 212 × 212 μm. The depth step along the Z axis was 0.5 μm with the number of slices being up to 200. For the experiments, 100% glycerol was used as the immersion medium. The image stacks were obtained for both sides of the scaffold samples, namely, the side adjacent to the collector electrode (bottom side) and the side facing the air during fabrication (upper side).

Processing, reconstruction, visualization, and analysis of the three-dimensional data arrays were performed using Avizo v9.0.1 software. (Thermo Fisher Scientific, Hillsboro, OR, USA). A detailed description is given in the article by Mikhutkin et al. [31].

### 2.4. Animals

B10.GFP mice expressing green fluorescent protein (GFP) and obtained from crossing the C57BL/10SnY and C57BL/6-TgN (ACTbEGFP) lines were used for these studies. The animals were obtained through self-breeding of the breeding stock, which was acquired from the State Research Center—Burnasyan Federal Medical Biophysical Center of the Federal Medical Biological Agency (Russia). All the mice were age-matched at six months and weighed 30 g. The animals were given water and food and were kept in the vivarium of the Russian Research Center for Molecular Diagnostics and Therapy (Russia).

### 2.5. The 30-Day-Old B10.GFP Isolation of GFP-Exp

The 30-day-old B10.GFP donor mice were euthanized via cervical dislocation. Complete tibias and femurs were extracted, and both ends of the metaphyses were removed. Sterile phosphate buffered saline (PBS) was slowly injected into the bone marrow (BM) cavity of the tibia, femur, and iliac crest to collect the BM cells. After isolation by the density centrifugation, the BM was plated in plastic tissue culture dishes in MEM Alpha Medium (Gibco, Thermo Fisher Scientific, Waltham, MA, USA) supplemented with 10% fetal bovine serum (FBS) (HyClone) and 1% gentamycin (Gibco, Thermo Fisher Scientific, Waltham, MA, USA). The media was changed 48 h after the initial plating to remove all nonadherent cells, and BM-MSCs were selected by plastic adherence. The cells were used after 3 to 6 passages for all the experiments. The adherent cells were detached by 0.05% trypsin–EDTA.

### 2.6. Characterization of GFP-Expressing Bone Marrow MSCs

Cell surface markers were quantified by flow cytometry after passage 6. The monoclonal anti-bodies included Sca-1, CD9, CD45, CD11b, and CD106 conjugated to Alexa Fluor 488 or phycoerythrin (PE) (Biolegend). For the measurements, 3 × 10^5^ cells were washed twice with PBS, resuspended in 100 µl of PBS containing monoclonal antibodies and incubated for 30 min at 40 °C. The cells were then washed twice and resuspended in 400 µl of PBS. A flow cytometer (BD FACSCalibur, BD Biosciences, San Jose, CA, USA) was used for fluorescent analysis. Nonspecific binding was determined using mouse IgG1-Alexa Fluor 488 and IgG1-PE negative controls. The percentage of positive cells was evaluated based on the fluorescence intensity.

Differentiation properties were also established after passage 6. For adipogenic differentiation, cells at a density of 5 × 10 cells/cm^2^ were cultured for three weeks in a 24-well plate in high glucose DMEM (Gibco, Thermo Fisher Scientific, Waltham, MA, USA) supplemented with 10% FBS, 0.5 mM isobutyl-methylxanthine (Sigma-Aldrich, St. Louis, MO, USA), 50 µM indomethacin (Sigma), 1 µM dexamethasone, and 5 µg/mL insulin. The induction medium was refreshed twice a week. Lipid droplets in the cell cultures were confirmed by Oil Red O staining. The cells were fixed in 2% formalin and washed in distilled water, followed by washing with 50% ethanol. For staining, a solution of 2% Oil Red O (Sigma-Aldrich, St. Louis, MO, USA) in 60% isopropanol was added and the cells were incubated for 15 min. After this step, the cells were washed with 50% ethanol and distilled water. The samples were examined under an optical microscope. 

For osteogenic differentiation, cells at a density of 3 × 10^4^ cells/cm^2^ were plated in α-MEM supplemented with 10% FBS, 10 mM β-glycerophosphate (Sigma), 50 µM ascobate-2-phosphate (Sigma), and 10^−9^ M dexamethasone (Sigma). The culture was maintained for three weeks. The medium was refreshed twice a week. Calcium deposition in the cultured cells was assessed by Alizarin Red S (ARS) staining. The cells cultured for three weeks were rinsed in PBS, fixed in 2% formalin, stained with 20 mg/mL ARS, and exposed for 30 min. The cells were then rinsed in water and washed again. Calcium deposition was examined under an optical microscope.

### 2.7. Cell Seeding of Fibrous PLLA Scaffolds

Three discs with 1.1 cm in diameter were cut from the 100 μm- and 600 µm-thick nonwoven PLLA mesh. The scaffolds were UV sterilized for 1.5 h in a laminar cabinet, washed in 70% ethanol, and then washed three times with sterile PBS. The scaffolds were incubated overnight in α-MEM, and then 5 × 10^4^ MSCs were seeded onto each PLLA scaffold in a 24-well tissue culture plate, followed by continued culture in the α-MEM. For in vitro analysis, the scaffolds were cultured for up to one week and the culture media was replaced every two days. For transplantation onto animals, these scaffolds were cultured for 7 days after being seeded with MSCs.

### 2.8. Cell Proliferation Assay

BM-MSC viability on the scaffolds was characterized by MTT (3-[4,5-dimethylthiazol-2-yl]-2,5-diphenyltetrazolium bromide) assays [32]. Briefly, after culturing in a 5% CO_2_ atmosphere and at 37 °C for 3 and 7 days, 20 μL of the MTT solution (5 mg/mL) was added to each culture well and the samples were incubated for 4 h. After the supernatant was removed, the formazan crystals were solubilized by 1 mL of DMSO. The absorbance at 490 nm was recorded by an automatic enzyme scanner (iMark, Biocompare, San Francisco, CA, USA). Cell survival was evaluated as the percentage of control. The control cells were detached by 0.05% trypsin–EDTA and the number of cells was counted.

### 2.9. ELISA Quantification of Cytokine Release

MSCs were seeded onto scaffolds or plated in 24-well plates (Corning Costar, Cambridge, MA, USA) at a concentration of 50,000 cells/well in α-MEM with 5% FBS. The conditioned media was collected after 72–168 h (5 specimens). VEGF, HGF, and TGF-β protein levels were quantified using the Quantikine ELISA Kit (R&D Systems, Minneapolis, MN, USA).

### 2.10. Fluorescence Imaging Methods

BM-MSCs seeded onto scaffolds for one week were fixed in 4% paraformaldehyde for 1.5 h. The cell nuclei were then stained with DAPI. Fluorescent images of MSC-seeded, 600 μm thick scaffolds were obtained using an Olympus Fluorview FV10i confocal scanning microscope (Olympus, Tokyo, Japan).

### 2.11. Animal Implantation Surgery

All of the animals were treated in accordance with the requirements of the Institutional Animal Experiment Committee at the Kurchatov Institute (Russia). Twenty-seven recipient C57BL/10SnY 6J mice (30 g, 6 months old) divided into nine groups of three mice each were used in the animal studies. Each mouse was anesthetized using ketamine (5 mg/kg). The 100 μm- and 600 μm-thick PLLA scaffolds with and without donor GFP-expressing BM-MSCs (5 × 10^4^ cells seeded onto scaffolds for one week) were implanted subcutaneously under the dorsal skin and were then allowed to develop and were biopsied in vivo over 21 days. At each of the post implantation timepoints (7, 14, and 21 days), the mice were euthanized, and the implants were individually dissected and removed from the subcutaneous dorsum.

### 2.12. Histological Analysis

Upon their removal from the subcutaneous dorsum, the implants were fixed with 4% formalin and embedded in paraffin. These specimens were sectioned (4 mm) along the longitudinal axis of the implant, and the sections were stained with hematoxylin and eosin (H&E).

### 2.13. Statistical Analysis

All the data are represented as the means ± standard deviation (SD) with three independent replicates. The differences between groups were analyzed by Student’s *t*-test. A *p*-value < 0.05 was considered to be statistically significant [33].

## 3. Results

### 3.1. SEM Study of the Scaffolds

The 100 μm- and 600 μm-thick PLLA 3D microfibrous scaffolds consisted of continuous fibers that varied in diameter size between 4 and 18 µm, with a mean diameter of 8.8 ± 1.3 μm. The secondary electrons (SE) SEM images of cross-sections of the 100 μm- and 600 μm-thick scaffolds are presented in Figure 1a,d, respectively. These and similar images demonstrate the dense microstructure of both scaffold tissues. The top side surface (facing air) (Figure 1b,e) appears less dense than the bottom sides (adjacent to the collector electrode) (Figure 1c,f). SE SEM images of the 600 μm-thick sample cross-sections sometimes demonstrated loose layers (see Figure 1d). 

We cannot rule out the possibility that the scaffold microstructure may be disturbed during cross-section preparation. Any other differences in the microstructures of the 100 μm- and 600 μm-thick samples were not revealed by SEM.

Since the SEM data did not allow to quantify the density (volume fraction) of the fibers and measure the pore space parameters, CLSM was used to determine these characteristics.

### 3.2. CLSM Study of the Scaffolds

The results of the 3D reconstructions obtained by CLSM for the top and bottom sides of both scaffold samples are presented in Figure 2a–d.

Relatively high fiber density precluded us from obtaining data for the scaffold layers deeper than 60–80 μm, therefore, the analysis was performed for scaffold volumes limited by a thickness of 50 μm from each side. The volume fractions of the fibers in the 100 µm- and 600 µm-thick scaffolds obtained for top and bottom sides are presented in Table 1.

The analysis of pore space and estimation of pore radii were performed through the skeletonization of the pore space [34]. Again, the studied volumes were limited by a thickness of 50 μm from each side. The visualization of the pore space skeletons displaying the local radius of the pores at every point of the central line of the skeleton is shown in Figure 3a–d. The corresponding pore radius distributions obtained through the evaluation of these results using the round shape pore approach are presented in Figure 4a,b. Each histogram demonstrates the normalized impact of a radius value at all the radii ranges. The mean pore radius for both samples, estimated from each side, are presented in Table 2.

Thus, the fiber density at a specific depth is not uniform, the porosity is higher on the top of the scaffold (the side facing air). The difference between the top and bottom portions is more noticeable in the 600 μm-thick scaffold (see Table 1). Surprisingly, the pore size distribution in the 100- μm-thick sample demonstrates a higher dispersion (see Figure 4 and Table 2), which could be important for cell adhesion. The density of pores with diameters more than 20 μm, which is comparable to the size of MSCs, is rather high (note that pore radii are shown in the histograms on Figure 4). Again, the density of these “big” pores is higher in the 100 μm-thick sample on the top side (air) of the scaffold. The pore size histogram attained for the parts of the scaffolds facing the electrode demonstrated a similar microstructure with a very close pore distribution.

The surface area of the scaffold fibers was also calculated (see Table 3) for the 100 μm- and 600 μm-thick samples. The fiber contact points were not taken into account. The lateral dimensions of the studied stacks were 212 μm × 212 μm and the heights of the stacks were 20 and 50 μm for the top and bottom sides, respectively.

These data again pointed to a denser fiber microstructure in the 600 μm-thick scaffold and indirectly indicates a three-dimensional, rather than layer-by-layer, growth mode of fibrous scaffolds. In the 600 μm-thick scaffold, the density of the top-down oriented fibers is higher.

### 3.3. Optical Microscopy Characterization of In Vitro BM-MSC Adhesion and Proliferation from the Bone Marrow of GFP-Transgenic Mice

After four passages, the isolated BM-MSCs exhibited a heterogeneous population of fibroblast-like cells (Figure 5a) and GFP fluorescence (Figure 5b). However, high levels of GFP expression could be detected in only approximately 10% of BM-MSCs derived from B10.GFP mice. Of note, a characteristic property of MSCs is that they are always a heterogeneous population. The stromal-vascular fraction was used in these studies, which is composed of a very low percentage of “true stem cells”, with the rest being progenitor cells.

When cultured in adipogenic or osteogenic medium, BM-MSCs differentiated into adipocytes or osteoblasts. Lipid droplet formation was detected by Oil Red O staining (Figure 5c) and bone matrix mineralization was detected by Alizarin red staining (Figure 5d).

Flow cytometry analysis revealed that the BM-MSCs derived from the BM of B10.GFP mice were characterized by the expression of Sca-1, CD9, CD106, and CD44 and the lack of CD45 and CD11b expression (Table 4).

These results suggest that, similar to typical MSCs isolated from the bone marrow of GFP-transgenic mice, these cells exhibited fibroblastic morphology with a CD106+ CD9+ Sca-1+ CD45- CD11b- phenotype and osteogenic and adipogenic differentiation potential.

We did not achieve 100% differentiation (which could be achieved by increasing the culture time) because for our study, it was only necessary to demonstrate that the cells were MSCs and that they are generally able to differentiate into adipocytes and osteoblasts. MSCs in our experiments were used to show how, when inoculated into the matrix, these cells change their morphology and functional activity (i.e., their proliferation and secretion of cytokines) to attract endogenous cells into the matrix after implantation.

We evaluated the cytotoxicity of in vitro cultured BM-MSCs to examine cell attachment to and proliferation on 100 μm- and 600 μm-thick PLLA 3D microfibrous scaffolds after 3 and 7 days (Figure 6a). The initial number of seeded BM-MSCs decreased on both the PLLA scaffolds after 3 days, while their total number on the 100 μm- and 600 μm-thick PLLA scaffolds was almost equal. The 600 μm-thick PLLA scaffold retained the highest cell viability during the one-week culture period, as opposed to the 100 μm-thick PLLA scaffold, which had the lowest cell viability. The number of BM-MSCs on the 600 μm-thick PLLA scaffolds was two times higher than that on the 100 μm-thick PLLA scaffolds after 7 days. The less dense microstructure of the 100 μm-thick scaffold might be preferable for cell adhesion and may therefore enhance cell proliferation during the initial period. However, the difference in the number of BM-MSCs after three days was negligible. Later, the MSCs penetrated the larger volumes of the 600 μm-thick PLLA, resulting in a greater level of cell proliferation. 

To steer the host response towards functional tissue regeneration through the design of scaffolds, it is important to identify a secreted growth factor. The release of growth factors by BM-MSCs after 3 days of incubation on 100 μm- and 600 μm-thick PLLA scaffolds and scaffold-free tissue culture plates (positive control) was significantly different (Figure 6b–d).

The level of HGF secreted after 3 days by BM-MSCs on the 600 μm-thick PLLA scaffolds was lower than that of the BM-MSCs on the 100 μm-thick PLLA scaffolds (2400 ± 600 and 13,585 ± 1000 pg/mL/million cells, respectively, *p* = 0.01), while the level of TGF-β was higher (7370 ± 245 and 4121 ± 25 pg/mL/million cells, respectively, *p* = 0.006). After 7 days, there was a significant increase in the level of HGF secretion by BM-MSCs on the 600 μm-thick PLLA scaffolds compared to those on the 100 μm-thick PLLA scaffolds (47,750 ± 2200 and 17,932 ± 1782 pg/mL/ppm, respectively, *p* = 0.02), although no differences were observed in TGF-β secretion at this timepoint.

Thus, both the 100 μm- and 600 μm-thick PLLA 3D microfibrous scaffolds served as biocompatible substrates for the proliferation and growth factor secretion of BM-MSCs. Again, the 600 μm-thick PLLA scaffold showed a higher efficiency due to a larger free volume, which resulted in better migration and adhesion of the cells into the inner layers.

### 3.4. The Effects of Various Structures and Properties of PLLA Scaffolds on Cell Morphology In Vitro Assessed by SEM

From a morphological perspective, variations in scaffold microstructure can affect cell shape. The cell morphology of BM-MSCs cultured for 3 and 7 days on 100 μm- and 600 μm-thick PLLA scaffolds was visualized by SEM (Figure 7). BM-MSCs stretched along the fibers of the 100 μm-thick PLLA scaffold after 3 days of culture, with a number of pseudopodia present (Figure 7a). By contrast, the majority of attached cells exhibited round cellular morphologies on the 600 μm-thick PLLA scaffold (Figure 7b).

After 7 days inside a 100 μm-thick PLLA scaffold, individual BM-MSCs exhibited a spindle shape (Figure 7c). In contrast, the 600 μm-thick PLLA scaffold provided less branching, but more spreading for BM-MSCs, which formed colonies (Figure 7d).

The analysis of the spreading and proliferation of BM-MSCs in vitro by fluorescence microscopy showed that, unlike cells cultivated in a tissue culture plate (Figure 8a), BM-MSCs cultured for 7 days in three-dimensional 600 μm-thick PLLA scaffolds formed colonies even at low seeding densities (Figure 8b,c). As was previously shown by Bauwens et al. [35], the proliferation of cells in the form of colonies is typical in most differentiation strategies and it produces subsets of cells with appropriate conditions for differentiation into specific cell types. The local microenvironment also modulates endogenous parameters that can be used to influence cell differentiation trajectories. Effective connections between colonies through cell–cell interactions, which are known to influence signal transduction, were observed in 600 μm-thick PLLA scaffolds (Figure 8d).

It should be noted that the differences in the sizes of cells represented in SEM images (Figure 7) and visualized by fluorescence microscopy (Figure 8) are seemingly due to different imaging conditions—SEM data were obtained in high vacuum, which resulted in partial shrinkage of cells and a decrease in their size. Moreover, all SEM images (Figure 7a–d) demonstrate cells in the scaffolds, whereas in fluorescence images, cells are more spread out in the culture plate (Figure 8a) or on the surface of the matrix (Figure 8b–d).

### 3.5. Influence of Microstructural Modifications of Scaffolds on Their Biodegradation after Implantation into Experimental Animals with and without BM-MSCs

To examine the influence of microstructural modifications of scaffolds as in vivo substrates for effective tissue growth, we subcutaneously implanted scaffolds without and with BM-MSCs into mice (Figure 9a). The implants were allowed to develop for up to 3 weeks in vivo and were excised and studied at 7, 14, and 21 days post implantation (Figure 9b). All of the resulting implants maintained their shapes after 1 week in vivo. However, after being cultured for 14 days, the surface area of the 100 μm-thick PLLA scaffolds decreased two-fold. In contrast, the 600 μm-thick PLLA scaffolds maintained an almost constant surface area, even after 14 days. At 21 days post implantation, the 600 μm-thick PLLA scaffolds with BM-MSCs were slightly decreased in diameter and thickness, while the same scaffolds without BM-MSCs did not change. By contrast, the 100 μm-thick PLLA scaffolds were resorbed and reduced to 0.5 cm in diameter after 21 days. The edges of these implants looked ragged. Thin fibrous capsules formed around the surface of all the scaffolds over time. 

### 3.6. Influence of Topographic Modifications of Scaffolds on their Biocompatibility after Implantation into Experimental Animals with and without BM-MSCs

The in vivo biocompatibility of the 100 μm- and 600 μm-thick PLLA scaffolds with and without BM-MSCs was examined by performing histological H&E staining of the implants at 7, 14, and 21 days post implantation (Figure 9c). The extent of host cell infiltration was observed 7 days after implantation with 600 μm-thick PLLA scaffolds without BM-MSCs. In contrast to this, intensive blood vessel formation and significant cell infiltration with the formation of their own extracellular matrices occurred with the 600 μm-thick PLLA scaffolds with BM-MSCs. For the 100 μm-thick PLLA scaffolds with BM-MSCs, the situation was different; an insignificant number of blood vessels and an uneven host tissue integration was observed 7 days after implantation. Moreover, hollow spaces were observed histologically in the implants, which were the cross-sections of the PLLA fibers.

The formation of blood vessels and significant host cell infiltration took place 14 days after implantation of the 600 μm-thick PLLA scaffolds without BM-MSCs, and the formation of hair follicles occurred with the 600 μm-thick PLLA scaffolds with BM-MSCs. Hair follicles also formed with the 100 μm-thick PLLA scaffolds with BM-MSCs, however, their appearance was not as clear as that with the 600 μm-thick scaffold. The scaffolds displayed a faster rate of degradation. 

Blood vessels formed 21 days after implantation of the 600 μm-thick PLLA scaffolds without BM-MSCs, and cell infiltration together with the formation of an intrinsic intercellular matrix were observed. After the same period of time, the 600 μm-thick PLLA with BM-MSCs was almost completely replaced by native issues. In contrast to this, with the 100 μm-thick PLLA with BM-MSCs, only a small number of hair follicles formed. Degraded PLLA fibers appeared as cavities lacking staining and they were more pronounced in the 100 μm-thick scaffold images.

Inspection of the images presented in Figure 9c indicated that inflammatory cells are present in the matrices. This is not surprising because the inflammatory phase is one of the stages in the regenerative process. However, in this study, our goal was to assess the possibility of matrix vascularization with a 600 μm-thick scaffold (i.e., a thickness greater than the diffusion limit (>200 μm)), as well as the integration of endogenous cells that form their own extracellular matrix. The newly formed vessels were clearly visible in the histological sections because of the endothelial lining, and erythrocytes are visible in the lumen of the vessel; hair follicles also have a characteristic structure and some of them are highlighted in Figure 9c. 

Staining with eosin (pink color) indicates the formation of intercellular matrix. In this case, segments of the degrading polylactide fibers (unstained fragments) are visible.

Enzyme-linked immunosorbent staining is certainly a more demonstrative method than histology, however, vascularization can be proven by histological sectioning. The sinus vessels are visible in the form of small cavities including sinuses or wide capillaries filled with blood or plasma.

To localize GFP and BM-MSCs in the 100 μm- and 600 μm-thick PLLA scaffolds 21 days after implantation, immunofluorescence microscopy was performed and the results are presented in Figure 10. Only a few GFP-positive cells were found in the BM-MSC-seeded 100 μm-thick scaffolds (Figure 10a) compared to the BM-MSCs-seeded 600 μm-thick scaffolds (Figure 10b). GFP and MSCs were predominantly localized deep in the scaffolds in both groups.

## 4. Discussion

The role of scaffold design and microstructure in the modulation of implant tissue interaction has become extremely important. Among various scaffolds developed for clinical applications, fibrous scaffolds produced from synthetic polymers using the electrospinning technique have been receiving increased attention because of their ease of fabrication and the ability to control their composition, together with their structural and functional properties. Synthetic fibers mimic the fibrous component of the extracellular matrix. The 3D reconstruction of 100 μm- and 600 μm-thick PLLA scaffolds revealed that the growth mode is not purely layer-by-layer. In the process of growth, the fibers bend in the vertical direction and, as a result, form denser structures. This may explain the high fiber density and slightly smaller pore size of 600 μm-thick scaffolds. Larger pores on the surface of 100 μm-thick scaffolds are better suited to MSC adhesion and consequently induce a higher secretion of HGF during the first 3 days. Later, cells penetrate the scaffold core and larger scaffold volumes (i.e., the 600 μm-thick scaffold) are preferred for cell proliferation and increased HGF values. The growing interest in electrospun materials arises from the fact that the scaffold microstructure modulates cell adhesion and viability in a cell type-dependent manner. Specifically, the microstructure amplifies certain biological responses such as contact guidance and differentiation [36]. In particular, our previous investigations on that topic demonstrated that the scaffold microstructure exhibited effects on the Ad-MSC phenotype (e.g., multipotency and osteogenic/adipogenic differentiation) and influenced on the proliferative activity of BM-MSCs [28]. However, to date, the effect of fibrous scaffold microstructures on the paracrine functions of cells and BM-MSC viability after delivery into a site of tissue injury is still not clear.

BM-MSCs cultured on various PLLA scaffolds demonstrated that the cells preferred to be attached and stretched along the fibers of both types of scaffolds after 3 days. However, cell colonies with similar round morphologies were found by SEM only in 600 μm-thick PLLA scaffolds. The viability after 3 days was approximately the same, but the 600 μm-thick PLLA scaffold demonstrated much better cytocompatibility after 7 days. The cells differed significantly in their morphology and the level of growth factor secretion when cultivated on 100 μm- and 600 μm-thick PLLA scaffolds. We assume that the BM-MSCs cultured on 600 μm-thick PLLA scaffolds penetrate inside the scaffold after 7 days and form colonies inside the large pores.

When fibrous matrices are used for transplantation into a lesion zone for the purpose of temporary structural support of regenerating tissues, such as bones, livers, and kidneys, their vascularization is necessary for the normal functioning of these tissue cells beyond the diffusion limited region (>200 μm) [23,37]. HGF can support the migration, proliferation, and differentiation of fibroblast resident cells and recruit progenitor cells to the site of inflammation. TGF-β is a pleiotropic cytokine that has an immunosuppressive effect, stimulates proliferation and differentiation, and regulates the production of its own extracellular matrix. These factors play a significant role in the regulation of the initial stages of repair. The introduction of MSCs into bioresorbable polymer matrices and the creation of a normal microenvironment for the secretion of the abovementioned growth factors contributed to more efficient vascularization and tissue integration in the case of the 600 µm-thick PLLA scaffold with MSCs compared to this matrix without MSCs.

It was observed that HGF levels and BM-MSC secretion was much lower with the 600 μm-thick PLLA scaffold specimen after 3 days. However, the level of TGF-β with the 600 µm-thick PLLA scaffold specimen was much higher than that with the 100 μm-thick PLLA scaffold specimen. This result might be indicative of the effects of the scaffold microstructure on the cells; upon contact with fibers, the cells responded through adhesion and cytoskeletal reorganization, which induced changes in the cell shapes resulting in the round cells on the 600 μm-thick PLLA scaffold and the fully stretched out cells on the 100 μm-thick PLLA scaffold, leading to a difference in the cytokine secretion levels. After 7 days, the observed significant increase in HGF secretion by BM-MSCs on the 600 μm-thick PLLA scaffolds could be associated with an increase in their cell number. Our observation suggests that the scaffold microstructure may be beneficial for maintaining the proliferation of BM-MSCs, as well as the relevant cytokine repertoire. 

Many studies have associated an increase in cellular infiltration with an increase in the pore size of fibrous matrices obtained by electrospinning [38,39,40,41]. The differences in the rates of matrix degradation, cellular infiltration, and the formation of microvessels and hair follicles revealed in the current study indicate that these parameters also depend on the thickness and microstructure of the scaffolds.

We showed, both in vitro and in vivo, that because of the larger volume of free space in the 600 μm-thick PLLA scaffolds, there was a higher number of cells and consequently, a higher number of BM-MSCs secreting growth factors in the 600 μm-thick scaffold, which was proven to be better suited to vascular invasion and active cell infiltration. After 21 days, a significant number of donor cells persisted in the bulk of the 600 μm-thick PLLA scaffolds, although the cells were unevenly distributed and clustered in certain areas. It is worth noting that other investigations demonstrated tissue formation only within an approximately 100 μm-thick layer at the surface, while the scaffold interior remained largely acellular [42,43,44,45,46]. However, in these previous works, the proliferation of cardiac myocytes was studied, which could affect the cell colony formation in those scenarios.

Our in vivo experiments demonstrated that 100 μm-thick PLLA scaffolds exhibited a higher rate of morphological changes relative to the 600 μm-thick PLLA scaffolds at 21 days post implantation. The smaller overall volume of those implants may be one of the reasons for this difference. We also cannot rule out the obviously weaker mechanical characteristics of the 100 μm-thick PLLA scaffolds and their contraction by cells [47,48]. Several studies have pointed out the sluggish degradation behavior of PLLA scaffolds both in vitro [49,50] and in vivo [50]. The difference in stem cell behavior in various scaffolds together with the dissimilar degradation rate of the implants allows for treatment diversity. Although the aim of implant surface modification is to enhance regenerative potential and wound healing, tuning the microstructure and architecture of biomaterials to promote suitable in vivo cellular responses could be an appealing strategy to improve tissue regeneration. Furthermore, our study suggested that the microstructure and architecture of the scaffolds influence cell–cell signaling and cell activity, which is followed by the secretion of cytokines.

## 5. Conclusions

The microstructures of 100 μm- and 600 μm-thick fibrous PLLA scaffolds were studied by SEM and CLSM. The SEM results did not reveal a difference in the fiber thicknesses between the samples. By contrast, the 3D image processing of the CLSM data demonstrated variations in the fiber density and pore sizes at different areas of the scaffolds and when comparing the two scaffolds. The pore sizes were larger in the areas that faced air during scaffold formation and smaller on the bottom portion of the scaffolds (the side facing the electrode). The differences in pore sizes and larger pore density were higher in the thinner (100 μm-thick) scaffolds, which were noticeable within the top areas of the scaffolds. The viability of BM-MSCs, both in vitro and in vivo, was studied. The 600 μm-thick scaffolds demonstrated a strong preference for in vitro cell proliferation and HGF secretion even after 7 days in comparison with the 100 μm-thick scaffolds. The histological H&E staining of in vivo BM-MSC-seeded 600 μm-thick scaffolds after implantation demonstrated much more pronounced neovascularization, cellular infiltration, and hair follicle formation relative to 100 μm-thick scaffolds. However, the degradation of 600 μm-thick scaffolds was less intensive than that of 100 μm-thick scaffolds. After 3 weeks, a significant number of donor cells persisted in the bulk of the 600 μm-thick PLLA scaffolds only, although the cells were clustered in certain areas. These results point to a certain acceptable thickness in three-dimensional scaffold architecture that modulates in vitro and in vivo cellular responses.

## Figures and Tables

**Figure 1 polymers-14-04352-f001:**
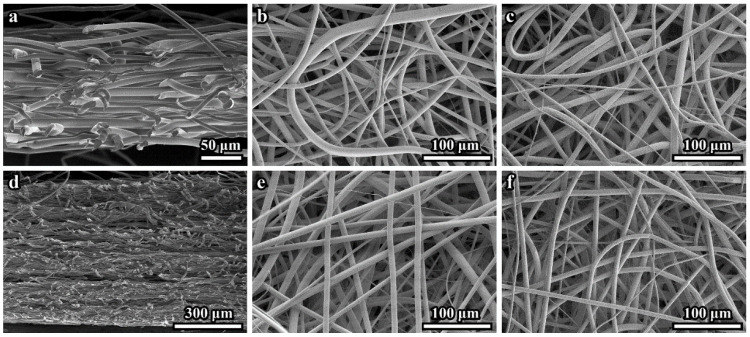
SEM images of the scaffolds: (**a**) cross-section of the 100 µm-thick scaffold; (**b**) planar view of the top (air) side of the 100 µm-thick scaffold; (**c**) planar view of the bottom (electrode) side of the 100 µm-thick scaffold; (**d**) cross-section of the 600 µm-thick scaffold; (**e**) planar view of the top side of the 600 µm-thick scaffold; (**f**) planar view of the bottom (electrode) side of the 600 µm-thick scaffold.

**Figure 2 polymers-14-04352-f002:**
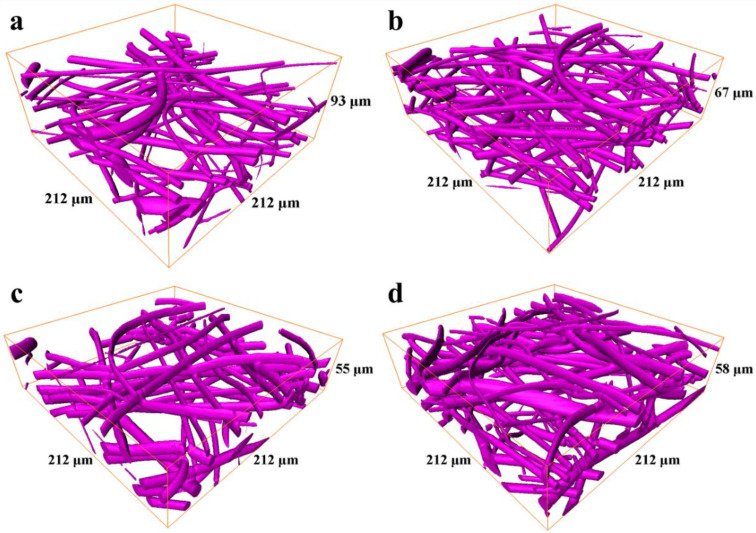
The results of the 3D reconstructions of the scaffold samples obtained by CLSM: (**a**) 100 µm-thick scaffold, top side; (**b**) 100 µm-thick scaffold, bottom side; (**c**) 600 µm-thick scaffold, top side; (**d**) 600 µm-thick scaffold, bottom side.

**Figure 3 polymers-14-04352-f003:**
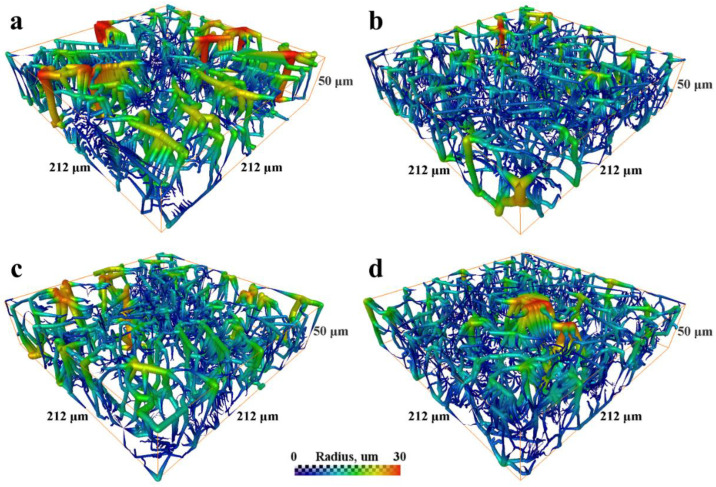
The visualization of the pore space skeleton (between PLLA fibers): (**a**) 100 µm-thick scaffold, top side; (**b**) 100 µm-thick scaffold, bottom side; (**c**) 600 µm-thick scaffold, top side; (**d**) 600 µm-thick scaffold, bottom side. The color and thickness of the lines correspond to the pore sizes.

**Figure 4 polymers-14-04352-f004:**
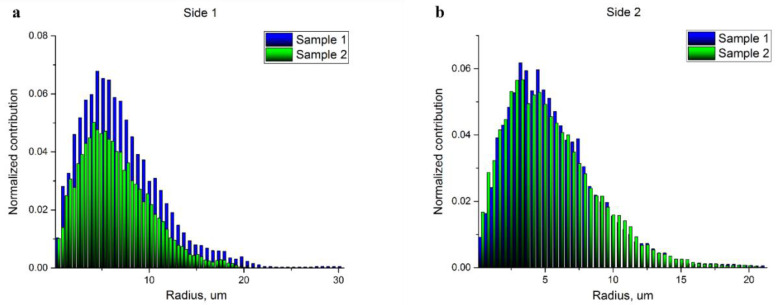
Histograms of the pore radius distributions: (**a**) top (air) sides of the 100 µm-thick scaffold (blue, sample 1) and the 600 µm-thick scaffold (green, sample 2); (**b**) bottom (electrode) sides of the 100 µm-thick scaffold (blue, sample 1) and the 600 µm-thick scaffold (green, sample 2).

**Figure 5 polymers-14-04352-f005:**
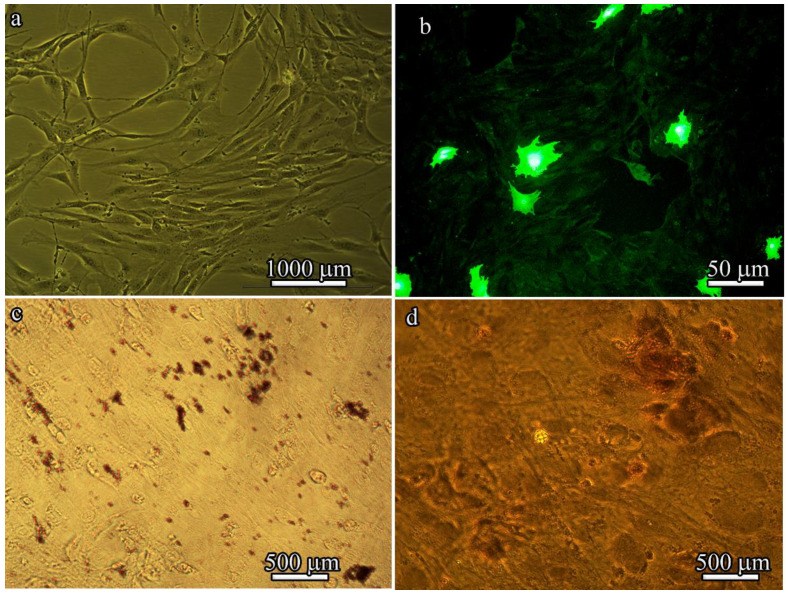
Characteristics of BM-MSCs derived from the bone marrow of B10.GFP mice. (**a**) Phase-contrast microscopy image of primary BM-MSC culture. (**b**) Fluorescent microscopy image of BM-MSCs. (**c**) Adipogenesis as indicated by neutral lipid vacuoles stained with oil red O on day 21 and imaged by light microscopy. (**d**) Osteogenesis as indicated by Alizarin red staining on day 21.

**Figure 6 polymers-14-04352-f006:**
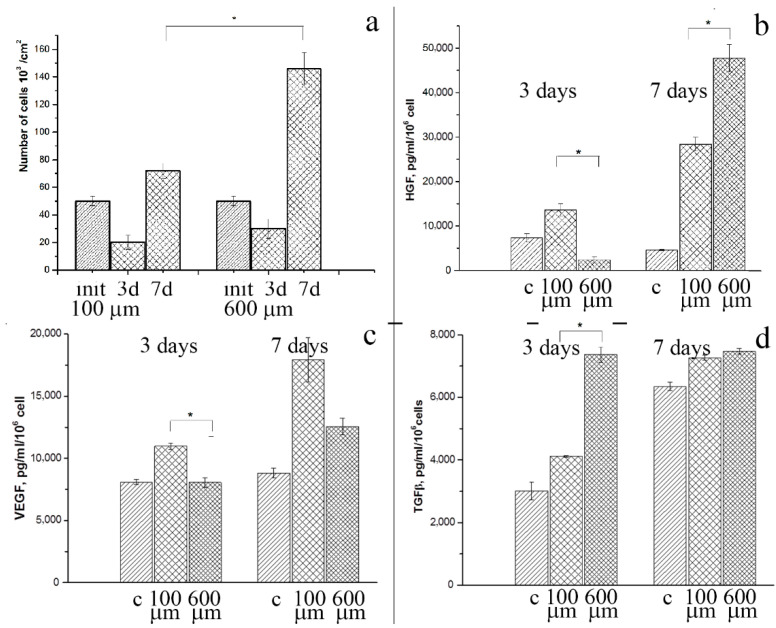
Biocompatibility of 100 μm- and 600 μm-thick PLLA scaffolds for BM-MSCs in vitro: (**a**) BM-MSC survival (init—initial number of cells); (**b**) HGF; (**c**) VEGF; and (**d**) TGF-β levels in conditioned media from BM-MSCs cultured on scaffolds compared to conditions media from plated BM-MSCs (c—control). Values shown are the means ± SD. * *p* < 0.05.

**Figure 7 polymers-14-04352-f007:**
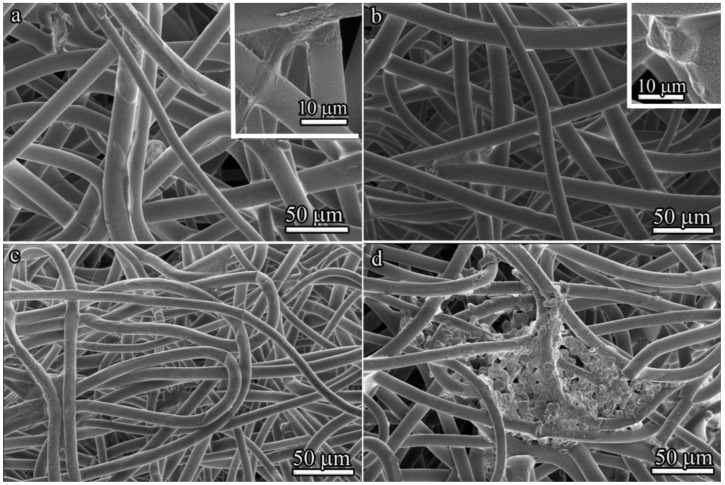
SEM images of BM-MSCs in PLLA scaffolds: (**a**) cells cultured for 3 days on the 100 μm-thick scaffold; the inset shows an enlarged scaffold fragment with an attached cell; (**b**) cells cultured for 3 days on the 600 μm-thick scaffold; the inset shows an enlarged scaffold fragment with an attached cell; (**c**) cells cultured for 7 days on the 100 μm-thick scaffold; and (**d**) cells cultured for 7 days on 600 μm-thick scaffolds.

**Figure 8 polymers-14-04352-f008:**
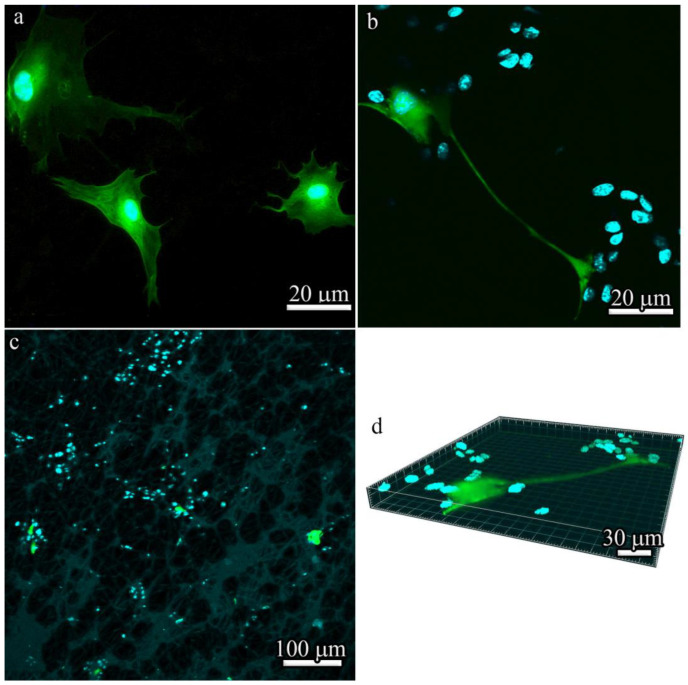
Fluorescence microscopy of BM-MSCs: (**a**) cells cultured in a tissue culture plate; and (**b**) cells cultured for 7 days on 600 μm-thick PLLA scaffolds. (**c**) Cells on the surface of 600 μm-thick PLLA scaffolds; (**d**) cells in 600 μm-thick PLLA scaffolds. Green areas represent GFP-positive cells, while blue areas represent DAPI-stained nuclei.

**Figure 9 polymers-14-04352-f009:**
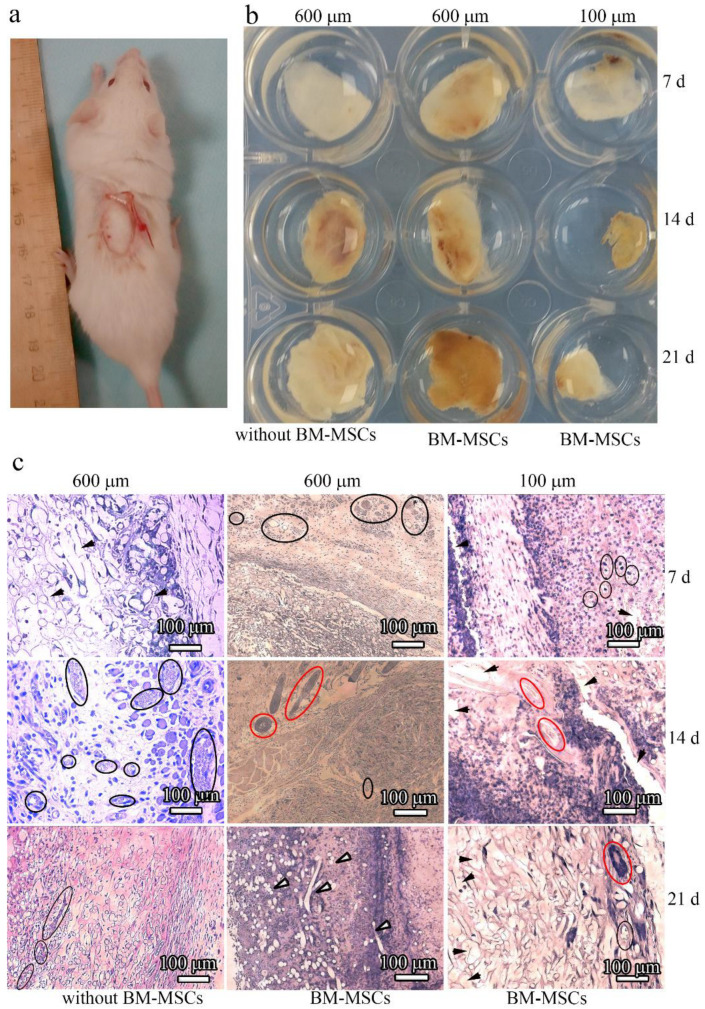
(**a**) The surgical procedure used to implant electroformed PLLA scaffolds. (**b**) Images of the 100 μm- and 600 μm-thick PLLA scaffolds grown with or without BM-MSCs upon removal from mice after 7, 14, and 21 days; (**c**) Images of H&E-stained in vivo implanted electrospun PLLA scaffolds grown with and without BM-MSCs (labeled at the bottom) after 7, 14, and 21 days (labeled on the right side). The PLLA thickness is marked at the top. The blood vessels are highlighted by black ellipses, the hair follicles by white ellipses, and degraded fibers by black arrows.

**Figure 10 polymers-14-04352-f010:**
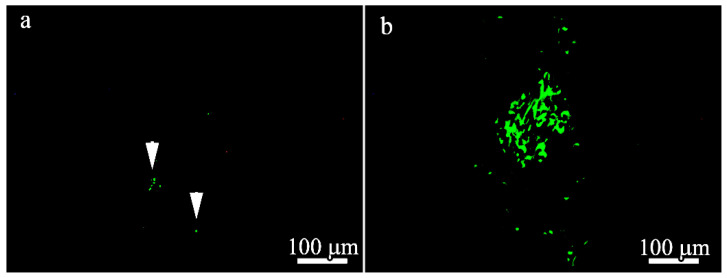
Fluorescent microscopy images of histologic sections 21 days after implantation of (**a**) 100 μm-thick (GFP-positive cells are arrowed) and (**b**) 600 μm thick PLLA scaffolds.

**Table 1 polymers-14-04352-t001:** The volume fraction of the fibers in 100 µm- and 600 µm-thick scaffolds.

Sample\Side	Top (Air)	Bottom (Electrode)
100 μm	7.3%	9.2%
600 μm	9.0%	12.2%

**Table 2 polymers-14-04352-t002:** Mean pore radii (with standard deviation).

Sample\Side	Top (Air)	Bottom (Electrode)
100 μm	7.1± 4.4 μm	5.6 ± 3.3 μm
600 μm	6.3± 3.7 μm	5.4 ± 3.4 μm

**Table 3 polymers-14-04352-t003:** The surface area of the fibers in the 100 μm- and 600 μm-thick scaffolds.

Sample	Stack Height	Side	
		Top (Air)	Bottom (Electrode)
100 μm	50 μm	116 × 10^3^ μm^2^	160 × 10^3^ μm^2^
	20 μm	33 × 10^3^ μm^2^	64 × 10^3^ μm^2^
600 μm	50 μm	146 × 10^3^ μm^2^	191 × 10^3^ μm^2^
	20 μm	53 × 10^3^ μm^2^	71 × 10^3^ μm^2^

**Table 4 polymers-14-04352-t004:** Identification of mouse MSC markers on BM-MSCs derived from the BM of B10.GFP mice.

Marker	The Average Fluorescence Intensity, MFI, a.u.	Percentage of Stained Cells, %
CD106	98	28.9%
Sca-1	164.4	95.7%
CD9	2465	96.6
CD45	2.4	0.5
CD11b	1.5	0.8

## Data Availability

The data presented in this study are available on request from the corresponding author.

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
