# Peer review of "The Performance of Nonwoven PLLA Scaffolds of Different Thickness for Stem Cells Seeding and Implantation"

_polymers, 2022, doi:10.3390/polym14204352_

Round 1
Reviewer 1 Report
The manuscript "On the Biocompatibility of Various Nonwoven PLLA Materials" has some interesting results. The main issues are that if reading the work it sounds unfinished.
1.The surface properties in view of roughness and pore sizes have been investigated before and plays a role in cell growth. Please add those different factors which can influence cell growth in the introduction
2. The main issue if using 100 μm PLLA and 600 μm why such gap why not a more analytical approach. Please add at least the middle thickness of 300 μm in it. It would be beneficial to know where the optimum of those thickness. How does a 900 μm PLLA behave?
3. The density of those scaffolds is another factor that should be investigated. In view of pore size the authors applied imagej software? How are real BET measurements? Would such not give a more clear picture about the surface area as well pore size?
4. How are the cell vitality and as seen from SEM images at 600 μm the cells more attached on 600 μm PLLA. How many cells survive a longer period?
5. The presentation of Figures need to be improved and reviewer suggest to create supplementary to move as example Figure 2 and Figure 3 can be transported to supplementary.
6. Did the authors made statistics and prove reproducibility of those samples? Are just one of each applied or several for each PLLA thickness?
7. Having such thick PLLA scaffolds where are the applications of such. please include it in the introduction.
8. Beneficial would be to add in the discussion a Table comparing cell grow factors on research made on scaffolds and what condition leading to enhanced cell growth in comparison to this work presented.
Author Response
Reviewer 1.
- The surface properties in view of roughness and pore sizes have been investigated before and plays a role in cell growth. Please add those different factors which can influence cell growth in the introduction.
Thanks for the comment. We have added the following text and corresponding references to the introduction (lines 76-84):
“The biocompatibility of scaffolds is determined not only by their composition, but also by free surface energy, electrostatic interaction, surface roughness, pore and fiber sizes [6-9]. Cell adhesion to nanofiber scaffolds (fiber diameter <1 µm) is higher than the adhesion to microfiber ones (fiber diameter >1 µm). However, smaller fiber diameter is often accompanied by insufficient scaffold pore size, preventing the cells from penetrating the scaffold [10]. On the other hand, while increased size of pores prevents the initial cell adhesion, it promotes the spreading of the cells and their differentiation. Changes in the fiber roughness do not significantly affect initial cell adhesion or differentiation, but can regulate the area of cell spreading.”
- The main issue if using 100 μm PLLA and 600 μm why such gap why not a more analytical approach. Please add at least the middle thickness of 300 μm in it. It would be beneficial to know where the optimum of those thickness. How does a 900 μm PLLA behave?
Thank you for the question and proposal. We changed the text (lines 457-464):
“These are the minimum and maximum possible thicknesses of the nonwoven materials. The lower limit of 100 μm is due to the minimum material strength and its ability to accumulate cellular structures. The upper limit is determined by the technological limitations of the electrospinning method. Preliminary study and our recent investigations of 400 μm PLLA nonwoven scaffold [27] demonstrated minor influence of thickness on the microstructure, thus in the present work we chose markedly different scaffolds.”
[27] Khramtsova, E.; Morokov, E.; Antipova, C.; Krasheninnikov, S.; Lukanina, K.; Grigoriev, T. How the Nonwoven Polymer Volume Microstructure Is Transformed under Tension in an Aqueous Environment. Polymers 2022, 14, 3526. https://doi.org/10.3390/polym14173526
- The density of those scaffolds is another factor that should be investigated. In view of pore size the authors applied imagej software? How are real BET measurements? Would such not give a more clear picture about the surface area as well pore size?
Thank you for the clarifying question. The density of the scaffolds (i.e. the volume fraction of the fibers, the ratio of PLLA fibers volume to the whole reconstructed 3D volume, including void space) and the sizes of pores were investigated using CLSM, and, as it is stated in the manuscript (lines 126-128) “Processing, reconstruction, visualization, and analysis of the three-dimensional data arrays were performed using Avizo v9.0.1 software (Thermo Fisher Scientific, USA). A detailed description is given in the article by Mikhutkin et al [31]”. ImageJ software was not applied.
While BET is the integral specific surface area measurement method that is very sensitive on nanoscale, it’s not always applicable in the analysis of microporous objects [30] and can give overestimated results compared to CLSM and SEM as it was shown in our previous work for surface structure estimation in the application for cells adhesion [31]. Since one of the main goals of our study was to compare the effect of the microstructural design of 3D fibrous PLLA scaffolds on cell adhesion and proliferation, the critical parameter was the size of pores between the fibers, rather than the nanopores on fibers’ surface, which makes BET measurements redundant. The combination of direct CLSM and SEM methods, on the other hand, is applicable on microscale and more appropriate for the studies of cell-matrix interaction.
The following text and corresponding references were added to the manuscript (lines 650-654):
“While BET is commonly used integral specific surface area measurement method that is very sensitive on nanoscale, it’s not always applicable in the analysis of microporous objects [29] and can give overestimated results [30]. Here for the 3D characterization of the scaffolds and analysis of its pore sizes the combination of direct CLSM and SEM methods was used.”
[29] Matthias Thommes, Chapter 15 - Textural Characterization of Zeolites and Ordered Mesoporous Materials by Physical Adsorption, Introduction to Zeolite Science and Practice. Editor(s): JiÅ™í ÄŒejka, Herman van Bekkum, Avelino Corma, Ferdi Schüth, Studies in Surface Science and Catalysis, Elsevier, Volume 168, 2007, Pages 495-XIII, ISSN 0167-2991, ISBN 9780444530639, https://doi.org/10.1016/S0167-2991(07)80803-2
[30] Yastremsky E.V., Patsaev T.D., Mikhutkin A.A., Sharikov R.V., Kamyshinsky R.A., Lukanina K.I., Sharikova N.A., Grigoriev T.E., Vasiliev A.L. Surface modification of biomedical scaffolds by plasma treatment // Crystallography Reports. 2022. V. 67. â„– 3. P. 421-427. DOI: 10.1134/S1063774522030233
- How are the cell vitality and as seen from SEM images at 600 μm the cells more attached on 600 μm PLLA. How many cells survive a longer period?
According to the experimental data from SEM images, after 7 days cells migrated inside the 600 μm PLLA matrix; therefore, under these conditions, for long-term cultivation of cells on such a matrix in vitro, it is already necessary to use the perfusion supply of the culture medium and transport CO2 and O2 to create optimal conditions. Also, dense colonies that formed by day 7 promote spontaneous differentiation of BM-MSCs, and our goal was to transplant proliferating stem cells. Therefore, we limited the cultivation of cells in vitro to this period.
- The presentation of Figures need to be improved and reviewer suggest to create supplementary to move as example Figure 2 and Figure 3 can be transported to supplementary.
Thanks for the proposal. However, we believe, that these Figures are important since we used them to estimate pore sizes and characterize pore space in general.
- Did the authors made statistics and prove reproducibility of those samples? Are just one of each applied or several for each PLLA thickness?
Thanks for the question. The number of samples for following cell seeding was added to the text (line 789):
“Three discs with 1.1 cm in diameter were cut from the 100 μm- and 600 µm-thick nonwoven PLLA mesh.”
We have also added the errors to the data in Table 2 (line 898-899).
- Having such thick PLLA scaffolds where are the applications of such. please include it in the introduction.
Done, thank you. Lines 463-465.
“It should be noted that relatively bulky scaffolds with a thickness of 600 μm could potentially be utilized to repair and regenerate injuries with stem cell co-culture for vascularization of implant.”
- Beneficial would be to add in the discussion a Table comparing cell grow factors on research made on scaffolds and what condition leading to enhanced cell growth in comparison to this work presented.
Thank you for the proposal. Growth factors expression by mesenchymal stem cells was studied for understanding functional activity of MSCs at the time of implantation of the scaffold with MSCs into experimental animal. High expression of these growth factors promotes vascularization and integration of the cells into the implant. This information is included in the "Discussion" section. We believe that it is not entirely possible to compare the expression of these factors with the relevant data from other studies, since cell sources, cell passages, methods of cell isolation, materials and structures of scaffolds differ.
Reviewer 2 Report
The title is very interesting regarding biocompatibility of nonwoven structures. I found the authors have put a big effort to collect information and results. However, the manuscript should be prepared based on the journal's requirements and as a research paper.
Additional comments:
1- Please change the title: On the Biocompatibility of Various Nonwoven Poly-L/L-lactide Materials
2- Improve the abstract based on the results and achievements.
3- Improve the introduction part. The novelty of the work should be emphasis. Please clarify the new idea of the work at the end of introduction.
4- Improve the results and discussion part.
5- Brief description on figures required.
6- Please use the journal template for the sections and the format of research paper.
Author Response
Reviewer 2.
The title is very interesting regarding biocompatibility of nonwoven structures. I found the authors have put a big effort to collect information and results. However, the manuscript should be prepared based on the journal's requirements and as a research paper.
We appreciate the comment very much. We rearranged the manuscript according to the journal requirements.
- Please change the title: On the Biocompatibility of Various Nonwoven Poly-L/L-lactide Materials.
Thanks again. Done.
- Improve the abstract based on the results and achievements.
We have added information to the abstract.
- Improve the introduction part. The novelty of the work should be emphasis. Please clarify the new idea of the work at the end of introduction.
We have rewritten the introduction.
- Improve the results and discussion part.
We tried to improve these sections.
- Brief description on figures required.
Thanks for the remark. We changed Figures 5, 9 and 10 and figure captions accordingly.
- Please use the journal template for the sections and the format of research paper.
Done.

Reviewer 3 Report
Comments
Title: On the Biocompatibility of Various Nonwoven PLLA Materials
Current research fabricated nanofibers from PLLA and evaluated the thickness on biocompatibility. The work presented in this article does not meet sufficient novelty and data of interest to the Polymers readers. PLLA is a well-known polymer have been used for various application specially drug delivery and tissue engineering.
For the other parts, the comments are:
1. Moderate English editing is required.
2. The lack of specific subject. Is it for bone regeneration or skin?
3. In figure 2, it seems there is no difference between 100 and 600 micrometer thickness in porosity.
4. If figure 5, poor quality of images specially for panel C and D.
5. If goal is bone regeneration osteogenic marker need to be studied well.
6. Scale bars for figure 5.
7. In figure 8 cells are bigger than 20 micrometers however in figure 7 panel D cells are smaller than 10 micrometer?!!!
8. How authors chosen the amount of nanofibers for implantation? By weight or area?
9. In figure 10 why the scale bare are different? Also, the quality of staining or imagining is poor. There is no similarity between pictures.
10. What is propose from showing figure 10?
Author Response
Reviewer 3
Current research fabricated nanofibers from PLLA and evaluated the thickness on biocompatibility. The work presented in this article does not meet sufficient novelty and data of interest to the Polymers readers. PLLA is a well-known polymer have been used for various application specially drug delivery and tissue engineering.
We absolutely agree with the reviewer that PLLA is a well-known polymer which have been used for numerous applications. However, we believe that to date tissue structures with a thickness of more than 200 µm (the limit of oxygen diffusion from nearby capillaries under in vivo conditions) have been studied insufficiently in vivo, which complicates its application for defect reconstruction in complex tissues. The reference [Capuana, E.; Lopresti, F.; Ceraulo, M.; La Carrubba, V. Poly-L-Lactic Acid (PLLA)-Based Biomaterials for RegenerativeMedicine: A Review on Processing and Applications. Polymers 2022, 14.1153] has been added to the manuscript (lines 91-92).
- Moderate English editing is required.
We tried to improve the manuscript.
- The lack of specific subject. Is it for bone regeneration or skin?
Thank you for your valuable comment. Indeed, we did not state the objectives of the study in the Introduction sufficiently clear. The aim of our study was not the differentiation of cells into any tissue. The scaffold was used for the implantation of proliferating MSCs to attract the body's own cells to the implant zone. The relevant sections have been modified accordingly.
- In figure 2, it seems there is no difference between 100 and 600 micrometer thickness in porosity.
That’s right. The difference in porosity between 100 µm and 600 µm scaffolds is faint and subjective. These images are mainly illustrative, thus we used CLSM in order to estimate the porosity parameters.
- If figure 5, poor quality of images specially for panel C and D.
We have changed Figure 5 c, d.
- If goal is bone regeneration osteogenic marker need to be studied well.
Osteogenic differentiation was not the goal of our work, but the use of these scaffolds to replace bone defects may be the topic of our next study and will certainly require the study of this marker.
- Scale bars for figure 5.
Thank you, done.
- In figure 8 cells are bigger than 20 micrometers however in figure 7 panel D cells are smaller than 10 micrometer?!!!
Indeed, cells visualized with fluorescence microscopy (Figure 8) appear bigger than the cells shown with SEM (Figure 7) due to different imaging conditions - SEM data was obtained in high vacuum mode, which led to the decrease in cell size. Moreover, all SEM images (Figure 7 A-D) demonstrate cells in the scaffolds, whereas in fluorescence images cells are more spread out in culture plate (Figure 8A) or on the surface of the matrix (Figure 8 B-D). Corresponding additions have been made to the manuscript (lines 1042-1047).
- How authors chosen the amount of nanofibers for implantation? By weight or area?
The amount of nanofibers for the implantation was chosen by area. In the recent paper by Rodina A.V., et. all. [“Migration and proliferative activity of mesenchymal stem cells in 3D polylactide scaffolds depends on cell seeding technique and collagen modification” Bull. Exp. Biol. Med. 2016, 162, 120–126] it was shown that the free volume of a scaffold with a thickness of 0.6 mm is 5.4x10-2 cm3, and with a thickness of 0.1 mm is 9x10-3 cm). If it is worth adding to the text, in your opinion, we are ready to do it.
- In figure 10 why the scale bare are different? Also, the quality of staining or imagining is poor. There is no similarity between pictures.
Thank you very much for the remark. We are sorry for the poor quality of Figures. We double checked the scale bars and changed them and several images in former Fig.10. In the new version of the text we combined Figure 9 and Figure 10. There was no similarity between the pictures, since the specimens were different in terms of thickness and the rate of scaffold degradation. Thus these differences are visible on histological sections.
- What is propose from showing figure 10?
In former Fig.10 (after the revision - Fig.9c) we tried to demonstrate that in addition to the scaffold degradation rate, the cellular composition differs. In the 600 µm scaffold with MSCs, cells migrate best, since the hair follicles was not detected at the corresponding time in other matrices. It can also be seen that the replacement of the 600 µm scaffold with its own tissue is more efficient. The 100µm scaffold with MSC degrades too quickly.

Round 2
Reviewer 1 Report
The authors made revision and answered all open question sufficiently
Reviewer 3 Report
Since the manuscript is suffering from the lack of novelty and specific subject I believe it is not worth to accept this manuscript in journal with IF 4.96.
At least authors could introduce a specific subject and design several experiments based on the subject for example bone regeneration.
But unfortunately, authors simply did electrospinning and evaluate the biocompatibility like research in 19th century.